# The Proteostasis Network: A Global Therapeutic Target for Neuroprotection after Spinal Cord Injury

**DOI:** 10.3390/cells11213339

**Published:** 2022-10-22

**Authors:** Scott R. Whittemore, Sujata Saraswat Ohri, Michael D. Forston, George Z. Wei, Michal Hetman

**Affiliations:** 1Kentucky Spinal Cord Injury Research Center, University of Louisville School of Medicine, 511 S. Floyd St., MDR616, Louisville, KY 40202, USA; 2Departments of Neurological Surgery, University of Louisville School of Medicine, 511 S. Floyd St., MDR616, Louisville, KY 40202, USA; 3Anatomical Sciences & Neurobiology, University of Louisville School of Medicine, 511 S. Floyd St., MDR616, Louisville, KY 40202, USA; 4Pharmacology & Toxicology, University of Louisville School of Medicine, 511 S. Floyd St., MDR616, Louisville, KY 40202, USA; 5MD/PhD Program, University of Louisville School of Medicine, 511 S. Floyd St., MDR616, Louisville, KY 40202, USA

**Keywords:** spinal cord injury, neurotrauma, proteostasis, ER stress, neuroprotection, cell death, white matter, oligodendrocytes, neurons

## Abstract

Proteostasis (protein homeostasis) is critical for cellular as well as organismal survival. It is strictly regulated by multiple conserved pathways including the ubiquitin-proteasome system, autophagy, the heat shock response, the integrated stress response, and the unfolded protein response. These overlapping proteostasis maintenance modules respond to various forms of cellular stress as well as organismal injury. While proteostasis restoration and ultimately organism survival is the main evolutionary driver of such a regulation, unresolved disruption of proteostasis may engage pro-apoptotic mediators of those pathways to eliminate defective cells. In this review, we discuss proteostasis contributions to the pathogenesis of traumatic spinal cord injury (SCI). Most published reports focused on the role of proteostasis networks in acute/sub-acute tissue damage post-SCI. Those reports reveal a complex picture with cell type- and/or proteostasis mediator-specific effects on loss of neurons and/or glia that often translate into the corresponding modulation of functional recovery. Effects of proteostasis networks on such phenomena as neuro-repair, post-injury plasticity, as well as systemic manifestations of SCI including dysregulation of the immune system, metabolism or cardiovascular function are currently understudied. However, as potential interventions that target the proteostasis networks are expected to impact many cell types across multiple organ systems that are compromised after SCI, such therapies could produce beneficial effects across the wide spectrum of highly variable human SCI.

## 1. Introduction

The pathophysiology of SCI involves the primary injury and a secondary injury cascade that progresses hours to months post-SCI [1,2]. There are 4 broad therapeutic approaches to spinal cord repair: (1) neuroprotection to prevent the progression of secondary cell death involving pharmacological or cellular therapies [3,4], (2) cell therapy to replace lost neurons or oligodendrocytes (OLs) [5], (3) promotion of ascending sensory, descending motor, and/or propriospinal neuronal regeneration and/or sprouting [6,7,8,9], and (4) enhancing neuroplasticity through rehabilitation/retraining [10,11]. Many of these have progressed to clinical trials, although none have proved substantively efficacious [12,13]. Preventing secondary cell death has great potential to therapeutically treat SCI patients. Historically, neuroprotective therapies have targeted single effector systems (e.g., specific ion channels, reactive oxygen species, and individual signaling pathways, to name but a few [3,4]) many of which affect only one type, or subtype, of neural cells. However, secondary neural cell death involves these and many other pathological processes, and to specifically target one, leaves the others unchecked. Moreover, multiple neural cell types undergo pathological insult after SCI and demonstrate an interaction between these post-SCI injured cells. A better approach would be to target more global secondary cell death mechanism(s) that occur in multiple neural cell types. In the past, hypothermia has been used in a similar therapeutic strategy [14,15,16] and currently, multiple prospective multi-center clinical trials in SCI are ongoing (www.clinicaltrials.gov as accessed on 20 October 2022) (see [17], for a recent discussion). Moreover, as cell death is a response to damage, interventions that promote the repair of such damage and restore cellular homeostasis may be needed for long-term tissue integrity and restoration of function. We contend that targeting the proteostasis network and its effector signaling pathways is a potential global therapeutic approach to facilitate neuroprotection in SCI.

The proteostasis (or protein homeostasis) network is defined as the “protein network with an immediate role in protein synthesis, folding, disaggregation, or degradation” [18]. This network includes all proteins necessary for translational, chaperone proteins needed for proper folding, as well as the ubiquitin proteasome system (UPS) and autophagy systems that degrade proteins (Figure 1). Proteostasis disruption activates stress response pathways including the heat shock response (HSR), the integrated stress response (ISR), the unfolded protein response (UPR), and/or the endoplasmic reticulum stress response (ERSR) [18,19,20]. These, often overlapping, stress responses determine whether proteostasis is restored or cell death is initiated (Figure 1). Likewise, stress-mediated disruption of proteostasis may activate autophagy and/or specific sub-pathways of the UPS [21,22,23,24]. In the following sections, we will discuss the current literature on the role of these proteostasis stress response pathways in the etiology of and recovery from SCI. This review covers data retrieved from PubMed based on search terms ‘spinal cord injury’ and the respective section headings. It is focused on preclinical studies of rodents, rabbits, and cats and does not include studies on lower vertebrates or invertebrates.

## 2. Overall Methodological Considerations

When interpreting the literature cited in this review, data must be critically evaluated as the validity of the conclusions that can be drawn depend entirely on experimental design and outcome measures evaluated. Some of the published literature in this field does not meet the necessary standards of experimental rigor. Outstanding collaborative papers have a detailed optimal experimental design for preclinical studies [27], as well as the types of data that should be collected for SCI studies and the ways in which they should be analyzed [28]. Interneuron or motor neuron loss after thoracic contusive SCI does not correlate with the extent of functional recovery [29,30]. Spared white matter (SWM) at the injury epecenter is the single variable that best correlates with the extent of functional recovery [31]. However, the therapeutic approaches we describe below are relevant to neuronal protection as well and may be therapeutically applicable to cervical and lumbar injuries, where neuronal protection is critical for recovery of function.

To assess changes in proteostasis effector molecules, techniques utilized include Western blots, immunohistochemistry, endpoint and quantitative polymerize chain reaction (PCR), various transcriptomic, lipidomic, and proteomic analyses, transgenic animals;both constitutive and conditional; and a range of imaging modalities. In his excellent review of synapse formation, Thomas Südhof presents a comprehensive discussion of the individual strengths and weaknesses of these methods [32] and the reader is referred to that discussion. Both gain (GOF) and loss (LOF) of function experiments are essential to conclusively ascribe a role for individual proteostasis effectors. Those GOF/LOF experiments can be genetic and/or pharmacological. Pharmacological enhancement or inhibition often suffers from a lack of drug specificity as off-target effects are widespread. This is a major issue with studies examining autophagy, which is discussed below in Section 4.2. The concern with global transgenic animals is that the targeted genes are affected in all tissues and there may be a compensatory mechanism(s) induced that substantively affect the systems being studied. Even conditional, tissue, or cell-specific deletion of genes has demonstrated off-target effects that must be considered [33]. Delineating a single effector in functional recovery after SCI is a very complex issue that requires careful experimental design and analysis. It is best to utilize multiple methods of analysis that take into consideration the caveats discussed above.

In each of the following sections, we define three categories of studies, each with increasing confidence in effector involvement in recovery from SCI. The first details changes in various proteostasis endpoint effectors with no attempt to examine behavioral recovery and/or define mechanisms. These are referenced but not extensively discussed as they provide little insight into physiological function. The second group consists of studies in which experimental manipulations of molecules that do not play a major primary role in proteostasis are shown to exert correlated effects on proteostasis and functional recovery. Some of these also histologically assess SWM. In the last group, GOF/LOF studies of direct proteostatic mediators with behavioral correlates (some with SWM analysis) provide the most reliable data on the involvement of the various aspects of the proteostasis network on functional recovery after SCI.

## 3. The UPS and SCI

The UPS is a highly regulated mechanism that degrades nearly 80% of all cellular proteins and is used to maintain intracellular protein homeostasis and turnover [26]. As such, the UPS serves as the main degradation pathway for misfolded proteins in protein quality control pathways that are employed by the nucleus, cytoplasm, and endoplasmic reticulum (ER) [24,34,35,36]. To ensure substrate specificity, only proteins linked to the poly-peptide cofactor ubiquitin (Ub), which also escape a de-ubiquitinating enzyme (DUB), are targeted to the UPS [26]. Proteins selected for UPS-mediated degradation are often short-lived proteins that are synthesized and degraded rapidly. They include regulatory proteins of cell cycle and apoptosis as well as transcription factors that mediate response to stress [37,38]. In response to cellular stresses such as oxidative damage, inflammation, or heat shock, the proteasome targets misfolded proteins with abnormal conformations with assistance from heat shock proteins (HSPs) that aid in identifying misfolded or unfolded proteins [39].

In the nucleus, increases in the burden of misfolded protein can impair cellular processes involved in DNA stability and replication, gene expression as well as ribosomal biogenesis [34]. To maintain nuclear protein homeostasis, the cell utilizes several UPS-mediated protein quality control degradation pathways and the nuclear proteasome [34]. Their disruption is associated with various forms of neurodegeneration [34]. In addition, UPS plays an important role in regulating the stability of proteins that participate in critical nuclear processes such as transcription [40].

The endoplasmic reticulum (ER) serves as a major cellular compartment for protein folding, sterol/lipid production, and free calcium storage. Given that one-third of all synthesized proteins in eukaryotic cells are channeled into the ER lumen destined for the secretory pathway, the UPS has an extraordinary role in maintaining ER homeostasis by contributing to the ER-associated protein degradation pathway (ERAD) [41]. In the ER, protein homeostasis can be compromised by pathological and/or physiological events that result in an imbalance between protein folding capacity and demand [41]. The ERAD pathway facilitates clearance of misfolded or unassembled protein substrates from the ER by the UPS and occurs in four steps: (1) recognition of aberrant ER proteins, (2) retrotranslocation into the cytosol, (3) polyubiquitination, and (4) proteolytic degradation by the 26S proteasome [24]. Moreover, misfolded proteins in the ER lumen, membrane, or cytosolic side are chosen for degradation through at least three distinct ERAD (ERAD-L, -M, and -C respectively) sub-pathways [36].

Contributions by the UPS to neuropathogenesis vary dependent on the type of pathology and/or UPS mediators. Proteasome inhibition has been shown to delay Wallerian degeneration in neurons derived from the sympathetic superior ganglia [42]. In rodent models of ischemia, proteasome inhibitors are neuroprotective by reducing infarct volume and attenuating the inflammatory response [43]. Conversely, the disruption of several ERAD components has been implicated in a number of neurodegenerative diseases that are characterized by the presence of misfolded protein aggregates including, but not limited to Huntington’s disease (HD) [44], amyotrophic lateral sclerosis (ALS) [45], and Alzheimer’s disease (AD) [46]. Interestingly, ERAD-mediated quality control of protein folding in the ER appears to be critical for OLs. OL-specific deficiency of the ERAD adaptor protein SEL1L (suppressor/Enhancer of Lin-12-like) led to the activation of PERK-mediated ISR [47]. Diminished myelin protein synthesis followed, resulting in myelin thinning in the CNS and later loss of OLs. PERK inactivation in double KO (*Perk* and *Sel1L*) mice rescues mature OL dysfunction, restores myelin protein translation and myelin thickness, and attenuates OL death when compared to *Sel1L* deficient mice [47].

The role of the UPS in modifying functional outcomes after SCI has been largely unexplored. After SCI, various cytotoxic events including hypoxia, hemorrhage, bioenergetics failure, oxidative stress, ER stress, and neuroinflammation contribute to secondary injury [1,2]. Oxidative stress, in particular, can lead to UPS dysfunction, decreased degradation of misfolded proteins, and proteotoxic stress [48]. After SCI, the accumulation of ubiquitinated proteins has been documented in both human and rat spinal cords, consistent with UPS dysfunction [49,50,51]. Altered expression of various UPS mediators has been documented both acutely and chronically (recently reviewed in [52]). However, the significance of those changes to the pathogenesis of SCI-associated tissue damage and functional recovery remains unclear. Given the critical role of ERAD for myelin maintenance and long-term OL survival, one can expect that neurotrauma-associated white matter damage may be enhanced if the UPS system is compromised in OLs.

Although still incompletely understood, the disruption of UPS-dependent protein homeostasis is a hallmark feature and contributor to not only SCI pathogenesis but also to several neurodegenerative disorders and therefore presents as a potential global therapeutic target.

## 4. Autophagy and SCI

Autophagy is an evolutionary conserved, catabolic, lysosomal degradation pathway required for cell homeostasis through the sequestration and breakdown of damaged or dysfunctional proteins and organelles [21,22,23]. There are three main forms of autophagy: macroautophagy, microautophagy, and chaperone-mediated autophagy (CMA). Macroautophagy (referred to here as autophagy) is the most well-studied form of autophagy and will be the focus of this review (Figure 2). Upon induction, dysfunctional proteins and organelles are isolated and packaged into the autophagosome, a double-membraned vesicle, which then fuses with a lysosome to form an autolysosome containing hydrolytic enzymes. Acidic conditions of the autolysosome lead to the degradation of the inner membrane and its cargo, and the broken-down contents are released into the cytosol and recycled by the cell. Autophagy is an adaptative response to cellular starvation. Under physiological conditions, autophagy is important for the clearance of misfolded proteins and damaged organelles to maintain cellular proteostasis. However, dysfunction or disruption of autophagy can have serious adverse implications, the end result often being programmed cell death. It is associated with many pathologies such as cancer, inflammatory diseases, and neurodegenerative diseases [21,22,23]. Due to its vital role in cell health, autophagy has become a promising therapeutic target in many pathological conditions.

### 4.1. Autophagic Pathway

Autophagy is active at basal levels, and under various conditions of stress (amino acid deprivation, hypoxia, oxidative stress, protein aggregation, ER stress) its activity fluctuates, often increasing, to support the needs of the cell [21,22,23]. Autophagy is controlled via a complex network of autophagy-related proteins (ATGs) (Figure 2). Those proteins together with the class III phosphatidylinositol 3-kinase (PI3K) regulate autophagosome initiation, formation, maturation, and its ultimate fusion with lysosomes to form autolysosomes. In the latter structures, lysosomal acidic hydrolases degrade the inner autophagosomal membrane and its cargo. Then, the degraded contents are released into the cytoplasm to be recycled by the cell. Adaptor proteins such as p62 recognize Ub chains on labeled proteins that can be selectively targeted for degradation via the autophagy pathway. A similar mechanism may also play a role in selective organelle autophagy. One should note that several autophagic regulators including Beclin1 or ATG5 or class III PI3K are required for that process and that their LOF produces selective disruption of that pathway. In addition, autophagy-associated post-translational modifications such as lipidation of LC3 provide a convenient marker of that process. For more details on the molecular biology of autophagy, readers are referred to excellent recent reviews [22,53].

### 4.2. Methodological Considerations

Autophagy is a complex process that requires careful experimental design for its evaluation. It can be monitored by direct observation of autophagic machinery or by quantification of protein and organelle degradation [54,55,56]. These two approaches measure steady state and autophagic flux, respectively. One can use volumetric morphometry/stereology (transmission electron microscopy—TEM) to measure steady-state levels of autophagic elements. However, caution is advised when using TEM as other cellular elements such as the ER, mitochondria, and endosomes, all of which contain a double membrane morphology can be confused with autophagosomes [56]. Other methods such as immunoblotting, flow cytometry, and fluorescence microscopy can be used to measure the status of autophagic machinery components such as LC3 and p62. However, without multiple time points and the use of lysosomal inhibitors, it is difficult to interpret these data with respect to autophagy function which is best assessed by determining autophagy flux.

Autophagic flux reflects the overall activity of the entire process of autophagy, which includes the formation of the phagophore, inclusion of cargo within the autophagosome, the delivery of cargo to lysosomes (via fusion of the latter with autophagosomes) and subsequent breakdown and release of the resulting macromolecules back into the cytosol. Measuring autophagic flux can be done by measuring LC3 in the absence and presence of autophagy inhibitors, and by assessing autophagy-dependent protein degradation [54,56]. Although not utilized as much as in the past, radiometric long-lived protein degradation assays remain one of the best approaches to measure endogenous protein degradation, such as autophagy [55]. At a minimum, lipidated LC3 (LC3-I and LC3-II) and p62 levels should be measured to assess autophagic activity because p62 directly binds to lipidated LC3 and is selectively degraded by autophagy [57,58]. Alternate powerful tools to measure autophagic flux utilize the biophysical properties of fluorescent proteins such as GFP and red fluorescent protein (RFP). The fact that GFP fluorescence is quenched in acidic environments such as the lysosome (pH 4–5) in contrast to RFP (or mCherry) offers the advantage to create an mRFP-GFP-LC3 tandem fusion protein reporter that can be utilized to measure multiple aspects of autophagy and autophagic flux both in vitro and in vivo [59,60,61,62]. If the autophagic flux is not measured, data on steady-state levels of various autophagy markers/mediators cannot be interpreted as evidence for either inhibition or activation of that pathway [56].

Importantly, the deletion of essential autophagy genes in mice (e.g., *Atg3*, *Atg5*, *Atg7*, or *Beclin-1*) results in embryonic and/or neonatal lethality [61,63]. To avoid the lethal phenotype, conditional tissue- or cell-specific deletion of essential autophagy genes can be applied to study the role of this process in physiology and disease [64,65]. Mice heterozygous for such LOF mutations have been also used with success [66].

Lastly, a cautionary note for pharmacological interventions that target the autophagic pathway must be taken into consideration. Drugs that enhance or block autophagy must be examined carefully due to the potential lack of specificity [56]. For instance, rapamycin enhances autophagy through the inhibition of mTORC1, but mTORC1 also acts as a master switch to regulate cell growth, translation, lipid synthesis, lysosomal biogenesis, energy metabolism, cell survival, and cytoskeletal organization [67]. Activators of the AMPK including resveratrol and metformin induce autophagy via AMPK-mediated inhibition of mTOR [68,69]. However, as the main sensor of cellular energy status, AMPK affects many more cellular pathways besides autophagy [70]. Use of inhibitors that target specific steps of the autophagy pathway such as 3-methyladenine (3-MA), chloroquine (CQ), and bafilomycin A1 (BFA) can help decipher whether autophagic flux has increased or decreased [54]. However, those drugs are not specific to autophagy and may become toxic due to the general disruption of cellular proteostasis. More specific autophagy inhibitors have been developed such as KU55933 and verteporfin, which target class III PI3K and autophagosome sequestration, respectively [71,72]. However, their specificity is not absolute and complications due to general disruption of the proteostasis are also expected. Relatively specific activators are also available including Tat-Beclin-1, a cell membrane-permeable peptide derived from a region of Beclin-1 protein, and a potent inducer of autophagy [73].

### 4.3. Status and Significance of the Autophagy Pathway after SCI

Since 2005, there are over 100 papers reporting studies of autophagy after experimental SCI including contusion, hemisection, transection, or ischemia specifically in animal models of rats, mice, or rabbits. The first group of the studies is descriptive and demonstrates changes in autophagy-related proteins after acute SCI with none extending past 14 days post-injury or assessing locomotor behavior, except to confirm injury severity [74,75,76,77,78,79,80,81,82,83,84,85,86,87,88,89]. The second group of studies utilized an intervention that either promoted or inhibited autophagy after SCI. While informative and insightful, the results from these studies are only correlative as the targeted effectors are indirect modifiers of autophagy. Many of these pharmacological interventions utilized non-specific drugs including metformin, rapamycin, or 3-MA that also affect other biological processes as discussed in Section 4.2. Moreover, such nonspecific approaches often resulted in conflicting conclusions regarding the role of autophagy in SCI pathogenesis. For instance, activation of autophagy with rapamycin was proposed to be neuroprotective and recovery-enhancing after thoracic contusions [90,91]. Others reported no effects of rapamycin following similar injuries [65]. Lastly, beneficial effects of inhibiting autophagy were proposed following unilateral cervical contusion in rats that were treated with bisperoxovandium to activate mTOR [92]. Therefore, we do not discuss those papers as their conclusions about the role of autophagy in SCI are correlative and likely obscured by additional effects of the interventions tested. Finally, three studies have applied the genetic LOF approach to induce selective disruption of autophagy and assess its consequences on pathogenesis and recovery after SCI [65,66,93]. As findings from those reports offer the most direct evaluation of autophagy’s role in SCI, they will be presented in Section 4.3.2.

#### 4.3.1. SCI-Associated Inhibition of Autophagy

Depending on the injury model and severity, as well as species and sex, increased expression of autophagy markers has been observed as early as hours to as late as 7 days post-injury [74,78,90,94]. Most observe accumulation of LC3-II, an indicator of autophagosomes, suggesting either induction of autophagy or perturbation of the autophagosome-lysosomal degradation pathway. The adaptor protein p62, which participates in selectively sequestering cargo for degradation, is also degraded in autophagosomes. Several studies report accumulation of p62 in spinal cord tissue after SCI regardless of injury type, severity, or species, and sex, supporting SCI-associated acute inhibition of autophagy [76,95,96,97]. As an expression of autophagy-initiating proteins did not change in most of those studies, increased levels of p62 and/or LC3-II suggest inhibition of autophagic flux after SCI.

Autophagic degradation requires fusion with the lysosome and the release of lysosomal proteases, such as the cathepsin family (i.e., CTSD), to degrade cargo. Lysosomal dysfunction has been reported in neurodegenerative diseases, CNS injuries, and aging, where lysosomal membrane permeability (LMP) is increased, causing leakage of cathepsins and other proteases, and neutralization of the acidic lysosome environment [98]. Recent findings suggest that cytosolic phospholipase A2 (cPLA2)-mediated damage of the lysosomal membrane contributes to the inhibition of autophagy after neurotrauma [99,100]. In the case of thoracic contusive SCI, the resulting accumulation of lysophospholipids led to lysosomal permeabilization, inhibition of autophagy, and exacerbation of neuronal loss [99]. Finally, one should note that SCI-mediated disruption of autophagy may result in the further collapse of proteostasis, including activation of the ERSR [65,95].

#### 4.3.2. Role of Autophagy in SCI: Insights from Autophagy LOF Mouse Mutants

Using OL-specific *Plp-cre^ERT2^*-mediated deletion of *Atg5* in mature OLs of adult mice, Saraswat Ohri et al. [65] showed greater myelin loss and restricted recovery of hindlimb function after T9 contusive SCI. Moreover, acutely after SCI, OLs in OL-*Atg5^−/−^* mice showed impaired autophagic flux and increased cell death. Such findings correlated with increased ER stress sensitivity of cultured *Atg5^−/−^* OL linage cells. In addition, partial general deficiency of *Atg5* in *Atg5^+/−^* heterozygous mice led to increased neuronal death acutely after thoracic contusive SCI [93]. Using a contusive thoracic SCI model in mice with Beclin-1 haploinsufficiency (*Beclin-1^+/−^*), Li et al. [66] showed that enhancing the SCI-associated autophagy inhibition in microglia/macrophages exacerbated neuroinflammation, white matter loss, neuronal death, axonal injury and persistent impairment of locomotion. Oral treatment with the autophagy activator, trechalose produced opposite effects. Thus, autophagy disruption follows spinal cord trauma and its further inhibition worsens tissue damage and limits recovery. Such effects reflect a critical, defensive role of autophagy in various types of spinal cord cells and suggest that selective enhancement of that process may be neuroprotective.

## 5. The HSR and SCI

Molecular chaperones are proteins that “facilitate the correct folding of newly synthesized proteins and refold proteins that have been denatured due to stress without becoming part of that protein’s final structure” [101]. Heat shock proteins (HSPs) are highly expressed proteins across all species and comprise 1–2% of all proteins in some cells [102,103]. They make up the majority of cellular chaperones and can be grouped into families based on their approximate molecular weight: HSP90, HSP70, DNAJ/HSP40, chaperonin/HSP60, and small HSP (sHSP) [101,104]. There are 15 and 4 mammalian homologs of HSP70 and HSP90, respectively, and they all interact with a large number of co-chaperone protein families to provide a vast array of cellular diversity. The DNAJ/HSP40 and sHSP co-chaperone families predominantly regulate HSP70. HSP90 chaperones include HSP27, HSP70-HSP90 organizing protein (HOP), cell division cycle 37 (Cdc37), p23, and activator of HSP90 ATPase (Aha1) [102]. They function in an ATP-dependent manner to reversibly bind to exposed hydrophobic regions of the protein, which is the non-native state of those proteins and facilitate their ultimate burial in the interior of the natively folded protein [101]. As cellular chaperone proteins are critical to the proper functioning of the proteostasis network (Figure 1), the HSPs play an important role in maintaining homeostasis in response to cellular stress.

HSPs have been most extensively studied in cancer (reviewed in [101,105,106]) where HSP90 was identified as a target of the antibiotic geldanamycin, a positive hit in an unbiased screen for anti-cancer drugs. It binds to the N-terminal ATP binding site of HSP90, as do the resorcinal analogs. Many structural modifications and small molecule mimetics have been developed from these backbone anticancer antibiotics, but none have proven successful in the clinical trials performed thus far. Newer drugs have targeted the C-terminal HSP90 domains, but have been equally ineffective clinically. The efficacy problems encountered deal with solubility and specific cancer targeting to eliminate off-target side effects. It is likely that once these drugs are better developed for cancer, they may find application in a variety of CNS neuropathologies, including SCI, to which HSP dysfunction has been linked [107,108,109]. However, as the goal of these drugs is to kill cancer cells and cell survival is paramount in CNS trauma, the therapeutic utility of inhibitors of HSR function to treat SCI is likely to be limited.

From a functional standpoint, the HSR has not been extensively studied in SCI. We identified 16 manuscripts that examined various aspects of the HSR in mice, rats, or rabbits after contusive, ischemic, or hemisection SCI. The majority of these were descriptive, showing how specific HSR parameter(s) changed after SCI. Increases in HSP70 and/or HSP72 mRNA and/or protein were seen acutely (<24 h) [110,111,112,113,114,115,116,117] or more chronically (7–42 days) post-SCI [111,112,118]. Conversely, Zhou et al. showed a slight drop in HSP90 acutely [119]. Definitive conclusions cannot be drawn from any of these studies regarding the role of those changes in HSP expression in mediating pathological and/or behavioral changes, as mechanism(s) were not addressed.

Another group of studies utilized interventions that caused changes in HSP expression and correlated those with accompanying effects on hindlimb locomotor recovery. While Chang et al. did not observe increases in HSP72 after SCI, pretreatment of the rats 5 days/week for 3 weeks with treadmill training exercise prior to injury resulted in enhanced locomotor recovery that was blocked by Hsp72 siRNA treatment [120]. Neuronal, and not white matter, sparing was correlated to behavioral recovery in this study, so it is difficult to conclusively interpret these results. Wang and Ren reported a significant reduction in HSP70 levels after contusive thoracic SCI which was partially reversed by treatment with allicin, an oily extract of garlic [121]. Concomitantly, allicin significantly increased hindlimb locomotor scores 7 days post-SCI. However, these authors also reported significant allicin-induced changes in catalase, superoxide dismutase, PI3 kinase, phosphoAkt, NFκB, and TNFα, so it is impossible to definitively ascribe changes in behavior to the altered HSP72 levels. Sharma et al. treated rats with a dorsolateral incision SCI (T10-T11) with a lipid peroxidase inhibitor H290/51 and reported it both blocked an injury-induced increase in HSP72 at 8 h post-SCI and slightly increased a 5 h Tarlov score [122]. However, no other proteostasis effector(s) was examined, and interpreting behavioral assessment that early after SCI is highly problematic due to animals still being in spinal shock [31,123]. None of these studies assessed lesion epicenter SWM preventing conclusive evaluation of those thoracic level SCI studies (see Section 2).

Finally, three different functional approaches were used to address the biological significance of HSR in SCI. Tanabe et al. gave matrine, a bacterial alkaloid that enhances HSP90 function, to mice with thoracic contusion SCI and showed an increase in hindlimb locomotor score as well as spinal cord 5-hydroxytryptamine (5HT) fiber sprouting [124]. In a second study, they showed that an HSP90 blocking mAb could partially abate the matrine-induced locomotor increase, which collectively shows the potential involvement of HSP90 in functional recovery after SCI. In lateral hemisection SCI studies in Hsp70.1^−/−^ mice, ipsilateral limb scores were slightly reduced and lesion volumes were increased [125]. However, there were methodological concerns with this report, as rat-specific BBB [123], rather than mouse-specific BMS [31] locomotor scoring was used, and SWM was not evaluated. Lastly, Klopstein et al. treated mice with α,β crystalline (CRYAB), a small HSP family member with structural similarity to HSP27, immediately after thoracic contusion SCI and showed an increase in hindlimb locomotor recovery one-month post-injury [126]. Interestingly, they also observed an immediate and sustained drop in CRYAB levels in OLs after SCI that was reversed by exogenous CRYAB, which correlated with an increase in dorsal column SWM and axon counts.

Collectively, these data, while by no means conclusive, support a role for HSPs in functional recovery after SCI. More specific GOF drugs need to be developed and utilized and both genetic and pharmacological studies need to be undertaken. It is likely that modulating HSP function will ultimately be one component of a successful, multifactorial therapeutic approach that targets both distinct and overlapping aspects of the proteostasis network.

## 6. The ISR/ERSR/UPR and SCI

The ISR, ERSR, and UPR comprise overlapping signaling modules that respond to a variety of stressors to initially attempt to restore cellular homeostasis and if unsuccessful, initiate cell death (Figure 3). ER stress response (ERSR) includes both UPR and ISR mediators (Figure 3). The ISR is associated with transient phosphorylation of the S51 residue of eIF2α (eukaryotic initiation factor 2α) by 4 distinct kinases (PERK, GCN2, PKR, HRI) which results in inhibition of general protein synthesis. Conversely, there is increased translation of specific stress-response mRNAs such as the ISR transcription factors Atf4 (activating transcription factor 4), Atf3 or Ddit3/Chop (C/EBP (CCAAT enhancer binding protein) homologous protein), chaperone proteins and ERSR/ER-associated degradation (ERAD) effector proteins [20,127,128]. PPP15RA/GADD34 (growth arrest and DNA damage gene 34), which is also among stress-induced proteins, is a regulatory subunit of PP1 (protein phosphatase 1) that recruits its catalytic domain to dephosphorylate peIF2α. Hence, PPP15RA mediates the negative feedback loop that switches off the ISR [129]. While various ISR kinases all converge on eIF2α as their main substrate, the spectrum of their upstream activating stimuli differs, albeit with overlaps [20]. PERK (ERSR-activated protein kinase RNA (PKR)-like kinase) is activated by ER stress, hypoxia-ischemia, oxidative stress, and oxygen-glucose deprivation. GCN2 (general control non-de-repressible 2) is activated by amino acid starvation, glucose deprivation, and UV irradiation. PKR (double-stranded RNA-activated protein kinase) is activated by viral infection (double-stranded RNA), oxidative stress, and ER stress. HRI (heme-regulated inhibitor kinase) is regulated by oxidative stress, iron deprivation, proteasome inhibition as well as cytosolic protein aggregation such as that during activation of the innate immune response.

Under normal homeostatic conditions, the three principal sensors/activators of the ERSR—PERK, ATF6, and inositol-requiring protein-1α (IRE1)—are bound to the chaperone protein GRP78/BiP (78 kDa glucose-regulated protein/binding immunoglobulin protein) [19,20,128,130]. After ER stress, GRP78/BiP disassociates from these proteins to facilitate increased ER luminal folding capacity. Concomitantly: (1) PERK dimerizes and autophosphorylates, activating itself to phosphorylate eIF2α as detailed above, (2) IRE1 alternatively splices X-box binding protein 1 (*Xbp1*) mRNA by excision of an intron that leads to a frameshift in its coding sequence. Then, XBP1s is translated and activates UPR target genes, and (3) ATF6 is transported to the Golgi apparatus where site 1 (S1P) and S2P proteases cleave it. Cleaved ATF6 translocates to the nucleus to activate UPR target gene expression. The ERSR IRE1/XBP1s and ATF6 signaling pathways make up the UPR and regulate the expression of various pro-homeostatic genes restoring the normal function of the ER [19,128]. Importantly, ERSR-activated PERK signaling overlaps with that of the other ISR kinases. The role of both signaling modules is to ameliorate protein damage in the ER and to restore homeostasis. If it cannot be restored, apoptosis is initiated.

There is a substantial functional overlap between the 3 arms of the ERSR as well as considerable compensation when one arm is genetically or pharmacologically deleted or inhibited. Utilizing K562 cells, Adamson et al. demonstrated both unique and overlapping gene expression programs for the 3 signaling arms of the ERSR, with PERK predominantly unique and IRE1 and ATF6 showing some functional redundancy [131]. However, IRE1 was the main driver of UPR responses to ER stress. In addition, LOF and GOF approaches showed both divergent and overlapping gene expression programs initiated by XBP1s and ATF6 [132,133].

One of the major effectors synthesized in response to ISR activation is the transcription factor ATF4. The ATF4-driven gene expression program includes many genes involved in the restoration of cellular homeostasis, including components of the anti-oxidant defense systems, amino acid synthesis, and translation. Upregulation of genes from the two latter categories may, with prolonged activation, also result in ATF4-dependent cytotoxicity which is further promoted by the ATF4-upregulated transcription factor CHOP [134]. In addition, CHOP increases the expression of pro-apoptotic genes including BH3-only members of the Bcl-2 family or death receptor-5 (*Dr5/Tnfrsf10b*) [130]. It also suppresses the expression of the anti-apoptotic gene *Bcl-2* [130]. Under oxidative stress, the ATF4-driven cytotoxic gene expression program is stimulated by HIF-PHDs (hypoxia-inducible factor-prolyl hydroxylase domain enzymes) [135]. Such ATF4 regulation can be targeted for neuroprotection with the CNS-permeable drug adaptaquin which attenuated ATF4-mediated gene expression and neurodegeneration in a mouse model of hemorrhagic stroke [135]. However, neither adaptaquin nor OL-specific deletion of *Phd1,2,3/Egln1,2,3* provides neuroprotection to enhance functional recovery in contusive thoracic SCI [136]. Thus, the cytotoxic ISR/ERSR is the dominant signaling pathway that drives chronic white matter damage and functional deficits. Accordingly, acute (<72 hr) changes in OL (*Olig2, Mbp*), neuronal (*Map2, Nse*), and ISR (*Chop, Atf4*) mRNAs can predict chronic (6 week post-injury) locomotor recovery after contusive SCI when the ISR/ERSR was modulated pharmacologically or genetically [137].

In evaluating the literature implicating the ISR/ERSR/UPR in the etiology of and functional recovery from SCI, multiple sexes, strains, and species of animals have been used. We analyzed 43 reports (2005–2021) that involved thoracic injuries (contusions, hemisections, ischemic injuries) in predominantly mice and rats, although a few rabbit studies are included. One group of studies simply shows changes in ISR/ERSR/UPR effectors secondary to SCI. Most examine acute changes occurring in the first few days, with none looking beyond 14 days post-SCI or assessing behavior recovery [138,139,140,141,142,143]. The details of these studies will not be highlighted as similar data have been described in more mechanistically insightful studies detailed below.

The second group of studies utilized an intervention to modify the ISR/ERSR/UPR acutely after SCI. The majority of these examined some aspects of behavioral recovery. The effectors used in these studies include necrostatin-1 [144,145]), lentiviral vector delivery of the ER stress-downregulated transcription factor ZBTB38 [146], LiCl [147], Di-3-n-butylphthalide (NBP) [148,149]), erythropoietin [150], amiloride [151,152], adenoviral vector delivery of prohibitin-1 [153], hyperbaric O_2_ [154], valproic acid [155], lentiviral deliver of shRNA against calcineurin regulator RCAN1 [156], chloroquine [157], fibroblast growth factor 2 (FGF2) [158,159], FGF22 [160], nerve growth factor (NGF) [161,162]), retinoic acid [163], the flavonoid plant extract loureirin B [164], the sesquiterpene plant extract β-elemene [165], sestrin2/hypoxia-inducible gene 95 [166], and the microRNA *miR-384-5p* [167]. These studies are all correlative in that initial SCI-induced changes in ERSR proteins/mRNAs were accompanied by locomotor deficit which was partially reversed by drug/effector treatment. While suggestive of a relationship between altered ERSR and functional recovery, these studies are neither mechanistic nor conclusive as the interventions tested do not specifically target ERSR effectors. Moreover, they may have other targets that have been implicated in recovery from various CNS injuries (as discussed in the above cited manuscripts).

The third category of studies used pharmacological and/or genetic approaches that directly target effectors in the ISR/ERSR/UPR pathway. These studies allow more definitive conclusions regarding the role of the ISR/ERSR/UPR in the recovery from SCI. In all cases, behavioral analyses mirrored the observed changes/abrogation of the ERSR. The strongest of these studies did both GOF/LOF approaches to address this question. These papers are detailed below. Tauroursodeoxycholic acid (TUDCA), an exogenous chemical chaperone used to facilitate protein folding and ameliorate ER stress, was initially used in hepatoprotection [168] but is effective in TBI [169] as well as models of neurodegenerative diseases [170]. Colak et al. [171] Zhang et al. [172], and Dong et al. [173] used TUDCA after T8,9 weight drop, clip compression, or contusion injuries in male Wistar rats, KM mice, or female SD rats, respectively. The former study showed reduced neuronal apoptosis 24 h post-SCI and functional improvement on post-injury days 1-5. Other studies observed an acute (3–7 days post-SCI) reduction of SCI-induced expression of *Grp78/*GRP78 and *Chop*/CHOP mRNAs/proteins and enhanced locomotor recovery between 5–14 days post-injury. Another exogenous chemical chaperone phenylbutyrate (PBA) was given acutely after T9 clip compression SCI in female Sprague Dawley rats and resulted in behavioral improvement at 14 days post-SCI that correlated with reduced expression of GRP78, CHOP, and cleaved caspase 3 (CC3) [174]. None of these studies assessed lesion epicenter SWM.

Using more ERSR-specific approaches, the roles of individual arms of the UPR have been investigated. No significant changes in post-SCI recovery were seen in *Atf6^−/−^* mice [175]. In studies using mice with a Nestin-Cre-driven deletion of *Xbp1* in neurons and macroglia, locomotor recovery was worse after T12 hemisection [176]. In addition, these mice also showed reduced axonal regeneration after sciatic nerve injury [177]. Consistent with those reports, OL/OPC-selective deletion of *Xbp1* is detrimental for locomotor recovery after contusive thoracic SCI, where chronic (6-weeks post-SCI) declines in both OPCs and OLs were also observed [178].

With respect to the PERK-eIF2α-ATF4-CHOP signaling pathway, direct evidence of ISR/ERSR involvement in recovery from mid-thoracic contusive [179] or lateral hemisection [176] SCI was shown in *Chop^−/−^* or *Atf4^−/−^* mice, respectively. In the former study, increases in acute (72 h post-SCI) expression levels of ISR/ERSR effector mRNAs and protein were reduced and chronic (6 week post-SCI) locomotor improvement was seen that was paralleled by increased epicenter SWM [179]. The latter study showed worse function in *Atf4^−/−^* mice, but the lesion was a T12 lateral hemisection. Current literature supports the role of ATF4 as a mediator of tissue damage in traumatic CNS injury [135,180,181]. Therefore its apparent beneficial activity in the hemisection SCI may reflect relatively moderate tissue damage after such a lesion and its dependence on plasticity for recovery. Indeed, ATF4 is a positive regulator of neural plasticity [182,183]. With a very severe T9 thoracic contusive SCI, global deletion of *Chop* does not result in enhanced functional recovery [184] suggesting that additional ISR/ERSR mediators may be involved in the secondary tissue damage or more likely ISR/ERSR (and/or the secondary damage, in general) plays a relatively minor role when the initial tissue damage is so extensive.

Saraswat Ohri et al. [185] showed enhanced locomotor activity and epicenter SWM in SCI mice treated acutely (0, 24, and 48 h post-SCI) with salubrinal, which prevents peIF2α dephosphorylation under both basal and ER stress conditions. In contrast, no effects on behavioral recovery were observed when the ER stress-inducible dephosphorylation of peIF2α was targeted pharmacologically with guanabenz or genetically using *Ppp1r15a/Gadd34^−/−^* mice [186]. PPP1R15A/GADD34 is the ISR-inducible regulatory subunit of PPP1 which recruits the catalytic subunit PPP1c to dephosphorylate peIF2α. Guanabenz selectively inhibits interactions between PPP1R15A/GADD34 and PPP1c without affecting the constitutive peIF2α phosphatase complex consisting of PPP1R15B/CReP (constitutive reverter of eIF2α phosphorylation) and PPP1c [187]. Salubrinal disrupts the PPP1c binding of either regulator. Therefore, temporally precise and substrate-specific inhibition of peIF2α dephosphorylation may be needed to reduce SCI-associated white matter damage. Alternatively, the PPP1R15B-PPP1c complex may be the critical target for white matter protection after SCI, perhaps by targeting a distinct pool of peIF2α either at a subcellular- or cell type-specific level.

Thus, the ISR appears as a promising target for neuroprotective therapies in SCI. While various ISR mediators may have a complex, time and dose-dependent role in SCI pathogenesis, determining their contributions to injury outcomes is critical to identify optimal neuroprotective targets.

## 7. Unanswered Questions

As data on the biological significance of UPS or HSR in SCI are non-existent or inconclusive, respectively, determining the role of those proteostasis systems in a cell type-specific context is a major issue to be addressed by future research. Likewise, further details on mechanistic aspects of the demonstrated autophagy contributions are to be determined. Below, we discussed several unanswered questions regarding the role of proteostasis networks in SCI. To better focus this discussion we paid particular attention to ERSR/ISR/UPR. However, the issues that are considered below may also apply to other proteostasis systems.

### 7.1. What Are the Triggers for Proteostasis Stress in SCI?

While numerous studies document SCI-associated activation of ISR/UPR in various types of spinal cord cells, the direct causes for such responses are not clear. As ER stress results from an imbalance between ER loading with native proteins and their correct folding, either increased synthesis of proteins that are transported to the ER or reduced function of the ER protein folding/glycosylation/transport capacity may lead to ER stress. SCI-associated damage of the ER is, therefore, one likely cause of ER stress. For instance, the reactive product of lipid peroxidation 4-hydroxy-nonenal that rapidly accumulates after contusive SCI [188] forms adducts with ER proteins and triggers ER stress in human endothelial cells in culture [189]. In addition, ROS may perturb ER Ca^2+^ homeostasis by inhibiting SERCA or increasing the opening of ER Ca^2+^ channels [190]. Other loss of function mechanisms of ER stress may include ischemia-related reduction in energy supply for maintenance of ER Ca^2+^ stores [191]. In addition, as ER stress is a component of the interconnected proteostasis network, any disruption of proteostasis may also lead to ER stress. Hence, the SCI-associated inhibition of UPS or autophagy is also a potential contributor to ER stress activation [65,95].

Those ER loss of function scenarios that either affect ER directly or trigger ER stress secondarily to perturbed proteostasis is expected during the immediate acute response to injury. Conversely, excessive loading of native proteins into the ER and subsequent ER stress may occur during the reactive glia-mediated repair response to spinal cord damage or in SCI-activated neuroinflammatory microglia and macrophages. Interestingly, the MS-relevant and SCI-upregulated cytokine interferon-gamma was also shown to induce ER stress in OLs during myelination [192]. Hence, neuroinflammation mediators may be yet another factor that contributes to SCI-associated ER stress. Finally, recent work in non-neuronal cell line systems has established that the ISR-ATF4 pathway is robustly activated by various forms of mitochondrial damage and that not a single eIF2α kinase is exclusively required for such a response [193]. As oxidative stress-related mitochondrial damage is well documented in SCI [194], one can expect its contribution to the activation of the ATF4-mediated ISR pathway after SCI.

### 7.2. What Determines the Neuroprotective or Deleterious Outcome of SCI-Associated Activation of ISR/ERSR/UPR?

A major question is what determines the fate of the spinal cord cells that undergo ISR/ERSR/UPR, as each of its branches engages both pro-homeostatic and cell death-promoting effector mechanisms. A likely possibility is that the duration of ISR/ERSR/UPR activation changes its outcomes from restoration of homeostasis to irreparable cell damage and cell death [130]. Thus, in the case of PERK, transient attenuation of protein synthesis by increasing pS51-eIF2α levels reduces the further accumulation of misfolded proteins in the ER but is not compatible with cell survival over a longer period of time [130]. Therefore, PERK-driven transcription factors ATF4 and CHOP stimulate expression of the pS51-eIF2α phosphatase subunit GADD34/PPP1R15A as well as many other proteins that contribute to protein synthesis recovery [134]. Those include components of the ribosome, translation factors, amino acid transporters, and aminoacyl-tRNA synthetases. If ER homeostasis has been restored, such a recovery of protein synthesis ensures the resumption of normal function. If ER damage has not been repaired and protein folding functions of the ER have not been restored, the ATF4/CHOP-driven recovery of protein synthesis leads to increased generation of ROS in the ER, mitochondrial damage, mitochondrial ROS generation, and cell death [134]. Interestingly, such a cytotoxic outcome of premature restoration of protein synthesis/secretory pathway was also shown in ER-stressed cells using an unbiased single-cell RNASeq analysis that was combined with CRISPR functional screening across the genome [195]. In that case, the PERK-peIF2α signaling increased translation of the transcription factor QRICH1 leading to the upregulation of pro-translational genes including those involved in ER-associated translation. Deletion of QRICH1 or those ER translation mediators reduced ER stress toxicity. The apoptosis/cell death-inducing mitochondrial damage would likely be a result of ER ROS-induced Ca^2+^ release from the ER and the subsequent Ca^2+^ overload of mitochondria [190]. In addition, CHOP-mediated regulation of various pro-apoptotic genes (see Section 6) would raise the death potential of cells that recover from ER stress to ensure that the ER homeostasis restoration challenge is efficient in identifying and eliminating cells with dysfunctional ER.

In the context of SCI, such concepts are supported by extensive upregulation of the pro-translational gene expression program that coincides with activation of the PERK-ATF4/CHOP signaling [179,196]. In addition, both oxidative stress and calcium-mediated mitochondrial damage are well documented in SCI and mitochondrial protection by anti-oxidants or uncouplers improves tissue sparing and locomotor function in contusive SCI in rodents [194,197,198,199,200]. Importantly, as mitochondrial damage by itself, is a major trigger for the ISR pathway via all ISR kinases [193], it is possible that it may serve as an amplifier/extender of the ISR. Thus, the initial, PERK-mediated ISR/ERSR would be pro-homeostatic. Then, dependent on the status of the mitochondria, the pro-homeostatic ISR would be terminated and cell survival would follow or sustained cytotoxic ISR will be activated leading to cell death. The existence of such an ISR amplification loop remains to be tested.

A timing mechanism has also been proposed to explain a switch between the pro-homeostatic and pro-death effects of the most conserved ERSR/UPR mediator IRE1 [130]. Thus, initially, the IRE1 RNAse activity would activate XBP1 to promote the restoration of homeostasis. However, persistent ER stress and the resulting chronic activation of the IRE1 RNAse activity would lead to the degradation of many mRNAs essential for ER function and microRNAs that support cell survival. The pro-apoptotic ASK1-JNK signaling may also be activated by IRE1 under such conditions [130]. As a result, ER stress is further enhanced and cell death follows. During persistent ERSR/UPR, increased oligomerization of IRE1 that promotes its autophosphorylation may underlie such a change in RNA substrates switching restoration of homeostasis in favor of induction of cell death in the IRE1 kinase activity-dependent manner [130,201]. However, beyond the pro-homeostatic processing of *Xbp1*, the role of IRE1 in the pathogenesis of SCI remains to be determined.

### 7.3. What Is the Role of Proteostasis in SCI-Associated Neuroinflammation?

In SCI, neuroinflammation that is mediated primarily by microglia and macrophages is a major modulator of the secondary injury as well as the later resolution of the SCI-associated tissue damage [202]. Moreover, after SCI, neuroinflammation may spread throughout the nervous system and potentially compromise such functions as memory or mood [203]. Among the core mechanisms of post-injury neuroinflammation is microglia/macrophage activation by pattern recognition molecules (PRMs) such as TLR4 as well as cytokines [202,204]; each of those neuroinflammatory activators may be modulated by the ERSR. In a mouse model of retinal ischemia, ER stress was essential for the induction of CXCL10 expression [205]. CXCL10 is a key microglia-activating cytokine that is produced by damaged neurons and genetic deletion of its microglial/macrophage receptor CXCR3 resulted in reduced neuronal death after retinal ischemia [205]. Similarly, the PERK-mediated ISR was required for the upregulation of neuroinflammatory cytokines by ER-stressed mouse astrocytes and subsequent activation of microglia [206]. Interestingly, haploinsufficiency of PERK prevented such a response without a loss of pS51-eIF2α-mediated translational attenuation that promoted cell survival [206]. Therefore, there appears to be a threshold of ISR/ERSR activation whose crossing would instigate neuroinflammation. However, TLR4-mediated neuroinflammatory responses may be attenuated if mild ER stress is present in microglia [207]. Conversely, in peripheral macrophages, the IRE1-XBP1 pathway was shown to contribute to TLR4- and TLR2-mediated production of pro-inflammatory cytokines [208]. That inflammatory response was further enhanced by ER stress [208]. NFkB, the key transcriptional regulator of innate immunity, is regulated by various mediators of ERSR and such a regulation likely underlies the crosstalk between the inflammatory response and the ERSR [209]. Moreover, the HRI-peIF2α-ATF4/ATF3 arm of the ISR facilitates inflammatory response in macrophages by promoting proteostasis during inflammatory activation. Specifically, the HRI-mediated ISR is required for efficient activation of cytokine expression by those PRMs whose downstream signaling is associated with cytosolic protein aggregation [210]. Thus, future studies are needed to clarify the role of the ISR/UPR in SCI-associated neuroinflammation as well as its demonstrated spread throughout the nervous system.

### 7.4. Does Persistent Disruption of Proteostasis Affect Post-SCI Plasticity That Supports Functional Recovery and/or Chronic Dysfunction?

After SCI, disruption of proteostasis including activation of the ISR/UPR may also affect other types of injury response than tissue damage. Noteworthy, prolonged activation of neuronal ISR/ERSR such as that reported in mouse models of AD or TBI has been proposed to interfere with synaptic plasticity and contribute to cognitive impairment [211,212]. Likewise, maladaptive plasticity including neuropathic pain may be facilitated by neuronal ISR [213,214]. Conversely, at least some ISR mediators such as ATF4 may be required for the structural plasticity of axons including their sprouting and regeneration [177]. Future experiments are needed to determine whether ISR/UPR may regulate the functional and/or structural plasticity of neuronal circuitries that support or compromise post-SCI recovery.

### 7.5. Does Disruption of Proteostasis at Organismal Level Contribute to the SCI-Associated Systemic Disease?

There is an increasing recognition that chronic SCI is associated with systemic pathologies that affect the immune response, metabolism, and cardiovascular system [215,216]. While SCI-associated dysregulation of autonomic neural control over lymphatic organs or the heart is an important contributor to such complications, additional mechanisms may also be at play. For instance, work in nematodes and mice has documented the existence of transcellular signaling of the IRE1-XBP1s-mediated UPR from the worm nervous system to the gut and mouse hypothalamic POMC neurons to the liver, respectively [217,218]. Such cell non-autonomous induction of the UPR has been proposed to pre-emptively increase the resistance of peripheral organs to stress including efficient handling of nutrients after feeding [209]. At least in worms, neurotransmitter release appears to participate in the trans-organ UPR [217]. Conversely, unresolved ER stress in hypothalamic neurons decreases leptin resistance promoting excessive food intake and metabolic dysregulation [209]. Therefore, one can wonder if pro-homeostatic trans-organ communication that is driven by the UPR and can be disrupted by unresolved ER stress may be compromised after SCI. In support of that notion, an increase in hypothalamic neuron ER stress markers that coincided with upregulated expression of inhibitors of leptin receptor signaling was reported at 4 weeks after T9 severe contusive SCI in mice [219]. It remains to be determined if similar evidence of unresolved ER stress is also present in the liver, the gut, or the pancreas of chronic SCI animals and whether hypothalamic activation of pro-homeostatic UPR signaling can attenuate such changes, increasing leptin and insulin sensitivity. As discussed above, if the adaptive ISR/UPR is overwhelmed by the continuous action of ER stressors or age-related decline in pro-homeostatic UPR, the maladaptive ERSR promotes apoptosis as well as harmful activation of innate immune responses [209]. The latter process, which may involve ERSR-associated activation of the key transcriptional regulator of innate immunity NFkB, engages intercellular communication and may spread ERSR across an affected organ or throughout the body [209]. Therefore, one can speculate that a general increase in innate immunity that has been associated with chronic SCI is facilitated by ERSR signaling in the CNS and/or peripheral organs such as the liver [215]. In addition, one could wonder how post-SCI maladaptation of the autonomic nervous system that may lead to autonomic dysreflexia affects organismal-level proteostasis [220].

## 8. Concluding Remarks

There is strong evidence that proteostasis disruption occurs in the spinal cord tissue following experimental SCI in rodents. Moreover, proteostasis impairment modulates spinal cord tissue damage and functional recovery. Importantly, such modulation is likely mediated by pleiotropic effects across various cell types and functional modalities. Those pleiotropic effects that are beyond a direct regulation of cell death/cell survival are the least studied. Yet, their therapeutic modification may have profound consequences on outcomes including enhancement of functional recovery as well as attenuation of the SCI-associated immunosuppression, metabolic syndrome, and/or neuropathic pain. While evidence for the role of SCI-associated proteostasis disruption in humans and/or other large animals is still missing, proteostasis mediators are interesting candidates for therapeutic targeting. Development of pharmacological tools to modulate the activity of those mediators as well as mechanistic research on their cellular/molecular effectors are both needed to progress toward testing the proteostasis hypothesis in the clinic.

## Figures and Tables

**Figure 1 cells-11-03339-f001:**
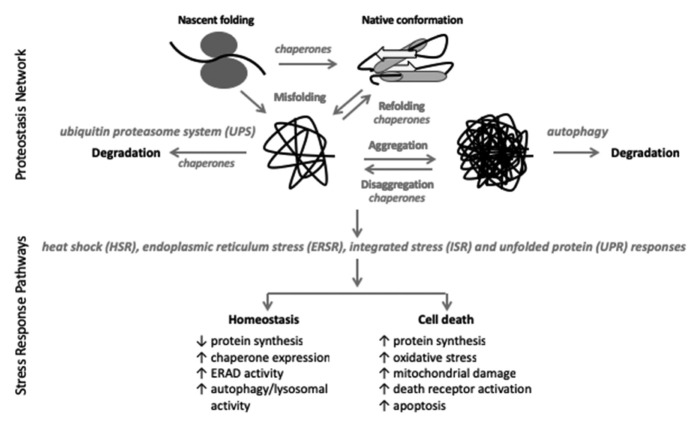
The proteostasis network (PN) consists of the proteins that synthesize, fold (chaperones), and degrade (UPS, autophagy) cellular proteins. The UPS degrades 90% of all cellular proteins and autophagy removes protein aggregates [25,26]. Proteins larger than 100 amino acids (~90% of all cellular proteins) do not spontaneously form their final native confirmation and require chaperone proteins to facilitate that process [18]. Despite chaperone involvement, under conditions of normally regulated proteostasis, 30% of newly synthesized proteins are immediately degraded; aging, stress, and cellular injury all further increase protein misfolding and degradation [18,25]. Integral to the PN is the stress response pathways including the HSR, ERSR, ISR, and UPR which are activated in response to cellular stress in an attempt to restore cellular homeostasis [18,19,20]. If cellular homeostasis cannot be restored, cell death programs are initiated [18,19,20].

**Figure 2 cells-11-03339-f002:**
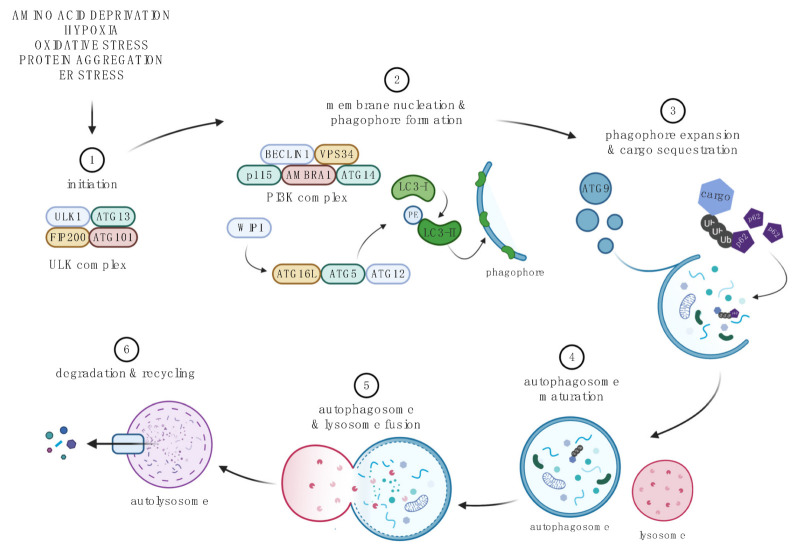
The macroautophagy pathway. 1. Initiation begins with the formation of the ULK complex, which is inhibited by mTORC1, and consists of ULK1, ATG13, RB1-inducible coiled-coil protein 1 (FIP200), and ATG101 [47,48,49]. 2. The PI3K complex, which comprises of Beclin-1, VPS34, AMBRA1, ATG14, and p115, together with the ULK complex, controls membrane nucleation and phagophore formation. WIPIs are recruited by phosphorylated lipids (PI3P), which in turn results in covalent conjugation of the ATG12-ATG5-ATG16L complex using ubiquitin-conjugation machinery. The ATG12-ATG5-ATG16L enhances the lipidation of LC3-I (Light Chain 3 (LC3) was originally identified as a subunit of microtubule-associated proteins 1A and 1B) to form LC3-II, conjugated to phosphatidylethanolamine (PE), and acts a scaffold to continue phagophore formation [47,48,49]. 3. ATG9-containing vesicles contribute to phagophore expansion. Cytoplasmic cargo such as aggregated proteins and proteins bound to p62 or LIR sites of LC3 is sequestered into the maturing autophagosome [47,48,49]. 4. Autophagosome is sealed, and SNARE proteins are recruited to form a mature autophagosome [47,48,49]. 5. Fusion with the lysosome releases protein-degrading hydrolases into the autophagosome. 6. Complete fusion results in the autolysosome. The inner membrane of the autophagosome and its cargo are degraded, and byproducts are released into the cytoplasm to be recycled by the cell [22,53]. Created with BioRender.com. For recent reviews on molecular mechanisms of autophagy see [22,53].

**Figure 3 cells-11-03339-f003:**
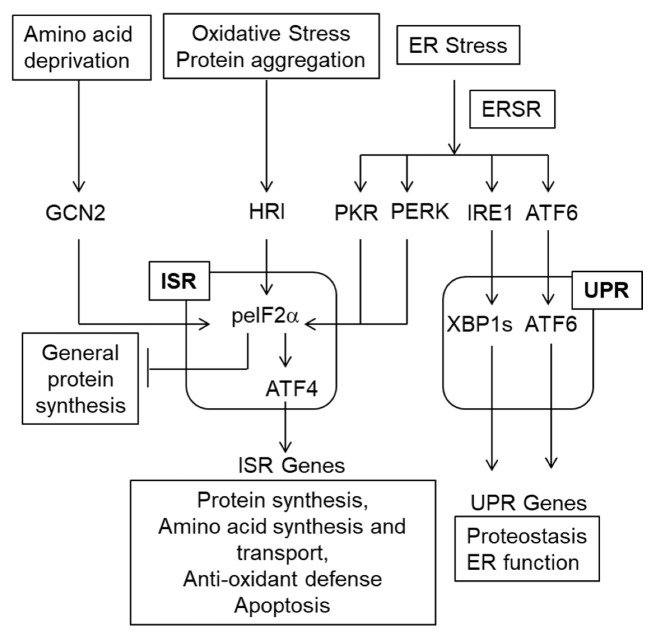
The ISR/ERSR/UPR pathway. Various stressors including ER stress activate the ISR [20]. The ERSR also includes the UPR [19]. Note that the ISR inhibits general protein synthesis and increases translation of select mRNA such as *Atf4*. Then, ATF4 mediates the transcriptional arm of the ISR. While regulation of gene transcription via sXBP1 and ATF6 is the major component of the UPR, IRE1 may also activate the pro-apoptotic kinase JNK as well as degrade ER-associated mRNAs as well as miRNAs (not shown on this schematic) [130]. Unlike the pro-homeostatic transcriptional UPR, those IRE1 activities are cytotoxic [130].

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
