# Peer review of "The Proteostasis Network: A Global Therapeutic Target for Neuroprotection after Spinal Cord Injury"

_cells, 2022, doi:10.3390/cells11213339_

Round 1

Reviewer 1 Report

This work summarizes the current knowledge about the role of the different effector systems of the proteostasis network on SCI, suggesting potential targets for new therapeutic approaches.

In overall it is an interesting review for researchers in the field 

Author Response

We appreciate reviewer's feedback. 

Reviewer 2 Report

In this paper, the authors review the literature of the cellular protein homeostasis networks in relation to Spinal Cord Injuries (SCIs). After explaining that effective therapeutic strategies to treat SCIs are missing, the authors introduce the therapeutic potential of targeting the proteostasis pathways. They then explain the different proteostasis pathways/networks and their potential impact in SCIs, followed by identifying unsolved questions and future directions.

The authors present an interesting and scientific sound review of the literature. The abstract and most of the paper is well written and easy to understand. However, some paragraphs do not fit in the flow of the current paper and are difficult to understand. Therefore, the review needs some structural improvements. In addition, it seems that although the main proteostasis pathways/networks are covered, the review does not balance between these. For example, the UPS is covered on page 4, but is missing from the section with unanswered questions.

Major points:

-It is unclear whether this review covers all literature concerning protein homeostasis and SCIs or concentrates on specific models such as rodents. Are zebrafish studies included?

-The authors discuss “pleiotropic effects” of the proteostasis network in SCIs. Although the reader can assume these pleiotropic effects, it would help to clearly state these and their consequences.

-The sentence on page 1 “Preventing secondary cell death has great potential to therapeutically treat SCI patients, although it will continue to be restricted to treatment of acute SCI.” is difficult to understand. What does the authors mean with the statement that preventing secondary cell death would be a restricted treatment to acute SCI? Please explain this in more detail.

-The last paragraph on page 2 is very wordy and difficult to read. It seems that the style is changing compared to the first 1.5 pages. It seems also that most information would fit better in an extended figure legend for Figure 1.

-Part 2 seems completely out of context. An introduction could maybe help the reader to understand why this part is here. Most of the first paragraphs is either too general or too specific. Restructuring and reworking the whole part 2 is necessary. What is exactly the goal of part 2? In the current version, this part of the review article destroys the flow without adding understandable information.

-Part 3 describes the UPS with a focus on ERAD. The nuclear UPS plays a pivotal role in many pathologies and is not mentioned here. Please add a paragraph about the nuclear UPS and its role in SCIs. If the nuclear UPS does not play a role or its role is currently unknow, is also worth describing. Furthermore, the UPS part is very short compared to the autophagy part. Is there less literature and less known? What about method to study the UPS in SCIs? The UPS is also missing from the unanswered question section. Are there not any unanswered questions in relation to the UPS?

-Part 5. The citation 97 seems to be very specific for a general description of molecular chaperones. Please use here a more general paper/review from the molecular chaperone literature.

-The second and third paragraph on page 10 would be improved by rewriting them with better transitions. The two paragraphs are very hard to read in the current version.

-The last paragraph of section 6 discussed unpublished data of the authors. I am not sure whether this is appropriate for this review. Please consider just using published data.

-The abbreviation WMS is not explained.

-Figures: Figure 1 has a very short figure legend. Please add a more explanatory legend. It is also unclear why the word Degradation is black and the words Misfolding, Refolding, Aggregation, and Disaggregation are red. What is the meaning of the color code? Figure 2 is very blurry. The figure legend is more detailed, but it seems that citations are missing in comparison to Figure 3 legend.

Minor points:  

-page 2, last paragraph: “This network includes the all proteins necessary for translational, chaperone proteins needed for proper folding, as well as the ubiquitin proteasome system (UPS) and autophagy systems that degrade proteins.” The first “the” should be deleted.

-page 3, 4th paragraph – suffers should be suffer

Author Response

Major points:

-It is unclear whether this review covers all literature concerning protein homeostasis and SCIs or concentrates on specific models such as rodents. Are zebrafish studies included?

Response: Added the following to the Introduction (p. 2, last paragraph):

<The review covers data retrieved from PubMed based on ‘spinal cord injury’ and the respective section headings. It is focused on preclinical studies of rodents, rabbits, and cats and does not includes studies on lower vertebrates or invertebrates.>

-The authors discuss “pleiotropic effects” of the proteostasis network in SCIs. Although the reader can assume these pleiotropic effects, it would help to clearly state these and their consequences.

Response: We modified the last sentence of the abstract as follows:

<However, as restoring proteostasis affects all cell types in all organ systems that are compromised after injury, effective therapeutic treatments could translate across the wide spectrum of highly variable human SCI.> 

-The sentence on page 1 “Preventing secondary cell death has great potential to therapeutically treat SCI patients, although it will continue to be restricted to treatment of acute SCI.” is difficult to understand. What does the authors mean with the statement that preventing secondary cell death would be a restricted treatment to acute SCI? Please explain this in more detail.

Response: We modified that sentence as follows:

<Preventing secondary cell death has great potential to therapeutically treat SCI patients.>

-The last paragraph on page 2 is very wordy and difficult to read. It seems that the style is changing compared to the first 1.5 pages. It seems also that most information would fit better in an extended figure legend for Figure 1.

Response: We have revised that section as requested.

-Part 2 seems completely out of context. An introduction could maybe help the reader to understand why this part is here. Most of the first paragraphs is either too general or too specific. Restructuring and reworking the whole part 2 is necessary. What is exactly the goal of part 2? In the current version, this part of the review article destroys the flow without adding understandable information.

Response:  1] The following sentences added at the beginning of the section:

<When interpreting the literature cited in this review, data must be critically evaluated as the validity of the conclusions that can be drawn depend entirely on experimental design and outcome measures evaluated. Some of the published literature in this field does not meet necessary standards of experimental rigor.>

2] The section has been considerably shortened, but feel free to additionally edit as you see fit.

-Part 3 describes the UPS with a focus on ERAD. The nuclear UPS plays a pivotal role in many pathologies and is not mentioned here. Please add a paragraph about the nuclear UPS and its role in SCIs. If the nuclear UPS does not play a role or its role is currently unknow, is also worth describing. Furthermore, the UPS part is very short compared to the autophagy part. Is there less literature and less known? What about method to study the UPS in SCIs? The UPS is also missing from the unanswered question section. Are there not any unanswered questions in relation to the UPS?

Response: We expanded the general information on UPS to include nuclear UPS. The SCI portion of the UPS part is short as there are no functional studies on that topic. We made that also clear in the short introduction that was added to Sec. 7 (unanswered questions):

<As data on biological significance of UPS or HSR in SCI are non-existent or inconclusive, respectively, determining role of those proteostasis systems in a cell type-specific context is a major issue to be addressed by future research. Likewise, further details on mechanistic aspects of the demonstrated autophagy contributions are to be determined. Below, we discussed several unanswered questions regarding role of proteostasis networks in SCI. To better focus this discussion we paid particular attention to ERSR/ISR/UPR. However, the issues that are considered below may also apply to other proteostasis systems.>

-Part 5. The citation 97 seems to be very specific for a general description of molecular chaperones. Please use here a more general paper/review from the molecular chaperone literature.

Response: We respectfully disagree with this comment. The cited reference (in R1 version it is reference #103) is taken verbatim from a major review from a leader in the field. We felt it appropriate to cite the source. Wikipedia defines molecular chaperones as “proteins that assist the conformational folding or unfolding of large proteins or macromolecular protein complexes” which could have been stated without reference. We believe it more appropriate to cite the source.

-The second and third paragraph on page 10 would be improved by rewriting them with better transitions. The two paragraphs are very hard to read in the current version.

Response: We have added transitions to those paragraphs as requested.

-The last paragraph of section 6 discussed unpublished data of the authors. I am not sure whether this is appropriate for this review. Please consider just using published data.

Response: All those unpublished data references have been deleted.

-The abbreviation WMS is not explained.

Response: Spared white matter (SWM) was defined in section 2. We inadvertently abbreviated it subsequently as WMS. Those have now all been changed to SWM.

-Figures: Figure 1 has a very short figure legend. Please add a more explanatory legend. It is also unclear why the word Degradation is black and the words Misfolding, Refolding, Aggregation, and Disaggregation are red. What is the meaning of the color code? Figure 2 is very blurry. The figure legend is more detailed, but it seems that citations are missing in comparison to Figure 3 legend.

Response: 1] The Figure 1 legend has been expanded.

2] The colors of the active processes have now been all changed to black.

3] Figure 2 resolution has been increased and its legend expanded.

Minor points:  

-page 2, last paragraph: “This network includes the all proteins necessary for translational, chaperone proteins needed for proper folding, as well as the ubiquitin proteasome system (UPS) and autophagy systems that degrade proteins.” The first “the” should be deleted.

Response: Modified as requested.

-page 3, 4th paragraph – suffers should be suffer

Response: Modified as requested.

Reviewer 3 Report

In this review; the authors presented a well-written article. The authors discussed proteostasis contributions to the pathogenesis of traumatic spinal cord injury (SCI). They contended that targeting the proteostasis network and its effector signalling pathways is a potential global therapeutic approach to facilitate neuroprotection in acute SCI. They have discussed the current literature on the role of the stress response pathways, the heat shock response (HSR), the endoplasmic reticulum stress response (ERSR), the integrated stress response (ISR), and the unfolded protein response (UPR), which collectively determine whether cellular homeostasis is restored or apoptosis is initiated, in the aetiology of and recovery from SCI. They have also described the therapeutic approaches relevant to neuronal protection.

The introduction, Figures, concluding remarks and References are framed correctly. Overall, the article is informative for the scientific fraternity. I recommend the paper for publication with below mentioned minor errors in the language and sentence framing.  

3. The UPS and SCI

3rd paragraph: Contributions by the UPS to neuropathogenesis vary dependent on pathology type,

4.2. Methodological Considerations

3rd Paragraph: conditional tissue or cell-specific deletion of essential autophagy genes can been applied

4.3. Status and significance of the autophagy pathway after SCI

proaches often resulted in conflicting conclusions as to the role of autophagy in pathogen

4.3.2. Role of autophagy in SCI: insights from autophagy LOF mouse mutants

Moreover, acutely after SCI, OL-Atg5-/- mice showed im[1]paired autophagic flux in OLs and increases in their SCI-associated death.

5. The HSR and SCI.

5th Paragraph:

served and an immediate and sustained drop in CRYAB levels in OLs after SCI that was

Author Response

 Reviewer #3: I recommend the paper for publication with below mentioned minor errors in the language and sentence framing.  

  1. The UPS and SCI

3rd paragraph: Contributions by the UPS to neuropathogenesis vary dependent on pathology type,

4.2. Methodological Considerations

3rd Paragraph: conditional tissue or cell-specific deletion of essential autophagy genes can been applied

4.3. Status and significance of the autophagy pathway after SCI

proaches often resulted in conflicting conclusions as to the role of autophagy in pathogen

4.3.2. Role of autophagy in SCI: insights from autophagy LOF mouse mutants

Moreover, acutely after SCI, OL-Atg5-/- mice showed im[1]paired autophagic flux in OLs and increases in their SCI-associated death.

  1. The HSR and SCI.

5th Paragraph:

served and an immediate and sustained drop in CRYAB levels in OLs after SCI that was

Response: All those suggestions were followed and manuscript was revised accordingly.

Round 2

Reviewer 2 Report

Thank you for this great revision! All my points are addressed